# Test and Simulation Analysis of Soybean Seed Throwing Process

**Dongxu Yan [1], Jianqun Yu [2], Na Zhang [2], Ye Tian [3] and Lei Wang [3,\*]**

1 Hua Lookeng Honors College, Changzhou University, Changzhou 213164, China
2 School of Biological and Agricultural Engineering, Jilin University, Changchun 130022, China
3 Center of Industry and Technology, Hebei Petroleum University of Technology, Chengde 067000, China
\* Correspondence: willa92@163.com; Tel.: +86-15803148001

**Abstract:** In order to analyze the effect of different factors on the bouncing and rolling distance of soybeans at the time of seed throwing, tests and discrete element method (DEM) are employed to analyze test soil and three representative soybean varieties. The parameters between soybean seed particles and soil particles are calibrated by means of a piling test and simulation. A seed throwing test apparatus is improved to analyze the effects of seed throwing height, soil plane inclination angle and collision orientation on the bouncing and rolling distance of soybean seeds. The effect of relative seed throwing speed on the bouncing and rolling distance of soybean seeds is analyzed using a computer vision seeding test bench. On this basis, the above-mentioned test procedure is simulated and compared with the test results. The results showed that the bouncing distance of the soybean seed particles was not significant. The rolling distance had a certain randomness when the seed throwing height was different. When the inclination of the soil plane became larger, the rolling distance increased. When the sphericity of the soybean seed particles was high, the effect of different collision orientations was not obvious. If the sphericity was low, the rolling distance was the shortest when colliding in the horizontal orientation and the longest when colliding in the vertical orientation. The larger the relative seed throwing speed, the larger the rolling distance of the soybean seed particles.

**Keywords:** soybean seed; soil; seed throwing; DEM; simulation; parameter calibration; bouncing; rolling

## 1. Introduction

In crop planting, it is crucial to achieve precision sowing. After sowing, seeds bounce and roll, which is an obvious reason to ensure precision seeding. The conventional test methods mostly directly measure the offset of seed particles relative to the drop position after seed drop, without exploring the influencing factors that produce bouncing and rolling. In this paper, a high-speed camera was used to design a seed throwing test device, and the influence of different factors on the seed bouncing and rolling distance was analyzed and studied in detail. The DEM was applied to the seed throwing test. The DEM simulation allows a better analysis of soybean-seed collisions from a microscopic perspective. At the same time, suggestions for optimizing the seed throwing process are suggested.

There is always a collision between the soybean seed particles and the soil particles during seed throwing. An accurate analysis of the influence of different factors on the bouncing distance (the maximum height reached by the seed after the first bounce) and rolling distance (the maximum distance measured from the point of impact) of soybean seed particles is essential. Bufton et al. [1] described the trajectory of different seeds after release from the metering mechanism through experimental tests. The seeds were made to impact the soil surface at a known impact velocity and angle so as to measure the amount of bouncing and rolling distance that occurred after the bounce. The test results showed that impact velocity and angle, soil surface properties and seed type all influenced the mean displacement and displacement variation. Ma Xu [2] used photoelectric sensors

to determine the seed landing speed and measured the bouncing and rolling distance of the seeds after landing in the seed groove. A mathematical model of the bounce and roll distance after seed bounce was further developed. A computer simulation was carried out by applying the mathematical model. The accuracy of the mathematical model was proved by comparing the simulation and test results. However, researchers have mainly used experimental methods to study the position of seeds after throwing and have tested fewer factors. Nowadays, the DEM is widely used in the field of agricultural engineering [3–8]. Therefore, the DEM can be combined with the test method to analyze in depth the influence of different factors on the bouncing and rolling distance of soybean seeds.

In the simulation analysis of the seed throwing test, the physical properties of seed particles and soil particles were different. The choice of contact model and how the parameters between the granular materials should be calibrated are less well studied. Hao et al. used the Hertz-Mindlin with JKR model to analyze the contact between hemp yam and soil particles, and calibrated the restitution coefficient, sliding friction coefficient and rolling friction coefficient between the particles by drop test, sliding test and rolling test, respectively [9]. Xu used the Hertz-Mindlin with JKR model to analyze the collision process between soybean and soil. The static and rolling friction coefficients were obtained by calibration. The restitution coefficient was measured by the collision of the particles with the soil disk. The surface energy between the soil particles and the soybean seed particles was taken from the surface energy between the soil particles [10]. Sui analyzed the collision process between soybean and soil particles by simulation. However, the accuracy of the contact model selection was not proved, and the parameters between soybean seed particles and soil particles were not calibrated [11]. The Hertz-Mindlin with JKR model [12–15] was chosen by all the above-mentioned scholars as the contact model for the analysis of granular materials with different properties. In the meantime, the aforementioned scholars calibrated the corresponding parameters without analyzing the sensitivity of the parameters, and such results are not accurate enough. Based on the above analysis the selection of the contact model, and the accuracy of the calibrated parameters, needs to be studied in depth.

Tian Yue Xu used a high-speed camera to study the bouncing and rolling of soybean seed particles after collision with an inclined soil plane. The process was analyzed by EDEM software. However, a high-speed camera cannot accurately record the movement of soybean seed particles in both the normal and tangential directions at the same time, which can lead to some bias in the results [10]. Meanwhile Xu only carried out a simulation of the collision between the soybean and the inclined plane, and did not analyze and study the effect of other factors on the bouncing and rolling of the soybean seed particles. Therefore, the seed throwing test apparatus needs to be further optimized, while the effect of different factors on the bouncing and rolling of soybeans needs to be studied in depth.

Based on the above issues, three representative soybean seed particles (SN42, with a sphericity of 94.78%; JD17, with a sphericity of 86.86%; and ZD39, with a sphericity of 80.6%) are used as the research objects in this paper [16,17]. The piling test is used as a calibration test, the sensitivity of the parameters is analyzed using the Plackett-Burman (PB) test and the sensitive parameters are calibrated using the central composite design (CCD) test. The effect of different factors on the bouncing and rolling of soybeans is further analyzed by improving the seed throwing test apparatus. The EDEM software is used to simulate and analyze the seed throwing process, showing the accuracy of the parameter calibration and the reliability of the simulated seed throwing test.

## 2. Contact Model

When using the DEM of simulation, the contact model and parameters have a significant influence on the simulation results. Therefore, before conducting the simulation of the seed throwing test, the contact model and the parameters of the simulation must first be determined.

The simulation analysis of the seed throwing test involves particles with two material properties, but only one contact model can be chosen between the particles. From previous

research conducted by the author [18], it is known that the contact model between soil particles is the Edinburgh Elasto-Plastic Adhesion (EEPA) model, so the contact model between soil particles and soybean particles should also be the EEPA model. The model is suitable for compressible, sticky or very sticky soil, soft and very sticky soil, materials such as clay and very wet sand [19–22]. The EEPA model is described as follows:

(1) Calculation of normal force

The total normal force $f_n$ in contact is the sum of the hysteresis spring force $f_{hys}$ and the normal damping force $f_{nd}$, and the formula is:

$$f_n = (f_{hys} + f_{nd})u \tag{1}$$

where, $u$ is the unit normal vector from the contact point to the center of mass and $f_{hys}$ is the normal contact force, which is related to the superimposed quantity by the formula:

$$f_{hys} = \begin{cases} f_0 + k_1\delta^n & k_2(\delta^n - \delta_p^n) \geq k_1\delta^n \\ f_0 + k_2(\delta^n - \delta_p^n) & k_1\delta^n > k_2(\delta^n - \delta_p^n) > -k_{adh}\delta^n \\ f_0 - k_{adh}\delta^n & -k_{adh}\delta^n \geq k_2(\delta^n - \delta_p^n) \end{cases} \tag{2}$$

$f_{nd}$ is the normal damping force, and its calculation formula is:

$$f_{nd} = \beta_n v_n \tag{3}$$

$v_n$ is the magnitude of the relative normal velocity, $\beta_n$ is the normal damping factor, and the formula is:

$$\beta_n = \sqrt{\frac{4m^*k_1}{1 + (\frac{\pi}{\ln e})^2}} \tag{4}$$

where, the recovery coefficient $e$ is the input parameter needed for the simulation, $m^*$ represents the equivalent mass of the particle, and its calculation formula is:

$$m^* = (m_i m_j / m_i + m_j) \tag{5}$$

where $m_i$ and $m_j$ are the masses of the particles.

(2) Calculation of tangential force

The contact tangential force $f_t$ is the sum of the tangential spring force $f_{ts}$ and the tangential damping force $f_{td}$, and the calculation formula is:

$$f_t = (f_{ts} + f_{td}) \tag{6}$$

The tangential spring force $f_{ts}$ can be expressed in incremental form, for which the calculation formula is:

$$f_{ts} = f_{ts(n-1)} + \Delta f_{ts} \tag{7}$$

where, $f_{ts(n-1)}$ is the tangential spring force of the previous time step, $\Delta f_{ts}$ is the increment of the tangential force, the calculation formula is:

$$\Delta f_{ts} = -k_t\delta_t \tag{8}$$

where $k_t$ is the tangential stiffness and $\delta_t$ is the increment of the tangential displacement, where $k_t = 2/7k_1$.

The tangential damping force $f_{td}$ is calculated as:

$$f_{td} = -\beta_t v_t \tag{9}$$

where, $v_t$ is the tangential relative velocity, $\beta_t$ is the tangential damping coefficient, the calculation formula is:

$$\beta_t = \sqrt{\frac{4m^*k_t}{1 + (\frac{\pi}{\ln e})^2}} \tag{10}$$

The limiting tangential friction $f_{ct}$ is calculated by applying the Coulomb friction criterion where the Coulomb friction criterion for the normal force is the normal force corrected by the adhesion force, and $f_{ct}$ is calculated as:

$$f_{ct} \leq \mu\left(\left|f_{hys} + k_{adh}\delta^n - f_0\right|\right) \tag{11}$$

where, $\mu$ is the friction coefficient. The total torque $\tau_i$ is calculated as:

$$\tau_i = -\mu_r \left|f_{hys}\right| R_i w_i \tag{12}$$

where $\mu_r$ is the coefficient of rolling friction, $R_i$ is the distance from the point of contact to the center of mass, $w_i$ is the unit angular velocity of the object at the point of contact.

A texture test was used to verify the applicability of the model, and the test device is shown in Figure 1a. Taking JD17 as an example, the test procedure was as follows. The soil tray was placed and the soybean particle fixed under the probe, then the texture analyzer was calibrated. The probe was moved downwards after starting the texture analyzer. After the soybean particle made contact with the soil, the force on the probe gradually increased. When the force reached its maximum, the moving probe started to move in the opposite direction until it stopped.

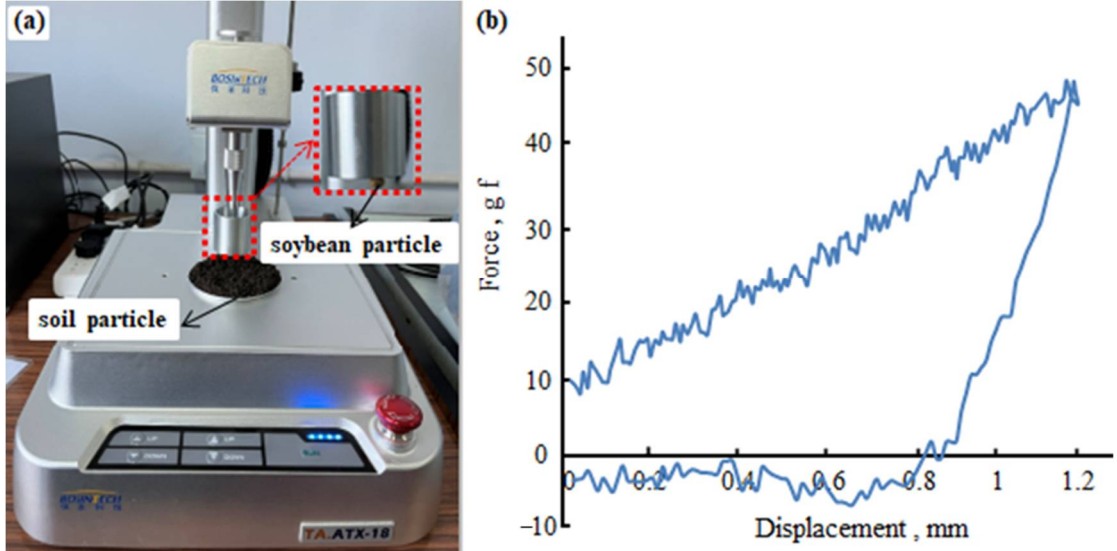

**Figure 1.** (**a**) Texture analyzer, and (**b**) the relationship between force and displacement of the probe.

The relationship between force and displacement of the probe is shown in Figure 1b. Analysis showed that when the displacement increased in the forward direction, the force gradually increased until it reached its maximum value. When the displacement reversely decreased, the force gradually decreased to zero, followed by the force reversely increasing to the maximum and then gradually decreasing until it reached a relatively stable state. The above test analysis demonstrated that the soil and soybean particle were compressible and sticky materials. Therefore, the EEPA model was applicable.

### 3. Simulation Parameters

The parameters between soil particles and between soybean seed particles were obtained in a previous study by the author [18], as shown in Table 1. Other initial simulation parameters of soybean seed particles are described in the authors' previous study [16–18].

**Table 1.** The parameters between soil particles and between soybean seed particles.

|  | Soil-Soil | SN42-SN42 | JD17-JD17 | ZD39-ZD39 |
|---|---|---|---|---|
| $e$ | 0.6 | 0.627 | 0.562 | 0.668 |
| $\mu_1$ | 0.9 | 0.2 | 0.2 | 0.2 |
| $\mu_2$ | 0.7 | 0.02 | 0.03 | 0.02 |
| F, N | 0 | 0 | 0 | 0 |
| $\gamma$, J/m$^2$ | 1 | 0 | 0 | 0 |
| $\lambda$ | 0.35 | 0.35 | 0.35 | 0.35 |
| $n_0$ | 1.5 | 1 | 1 | 1 |
| $n$ | 1 | 1 | 1 | 1 |
| $k$ | 0.67 | 0.67 | 0.67 | 0.67 |

For the particles' block, 200 seeds of each variety were taken and the length, width and thickness of soybean seed particles were measured to calculate the volume standard deviation. The particles' block was generated according to the volume normal distribution. The simulation parameters between soybean seed particles and soil particles were calibrated using a piling test.

### 3.1. Calibration Tests

The piling test apparatus was mainly composed of soil tray, loading box and insert plate, as shown in Figure 2. The test procedure was as follows. The test soil was configured and its moisture content was measured at 24.32%. The moisture content of the soil was kept constant during the test. The soil was loaded into the soil tray. Soybean particles were loaded into the loading box. Once the soybean particles had stabled, the insert plates were pulled out. The soybean seed particles flowed out of the loading box and formed an angle of repose, as shown in Figure 3. The angles of repose were processed using image processing software and three replicate trials were carried out for each variety. The angle of repose of the soybean seed particles were 19.79°, 20.32° and 20.96° for SN42, JD7 and ZD39, respectively.

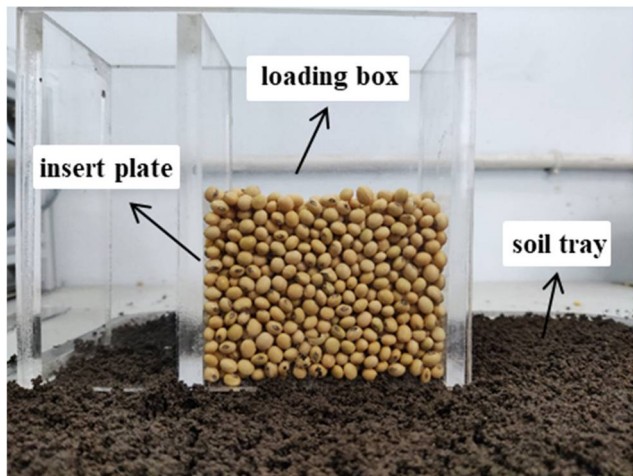

**Figure 2.** Piling test device.

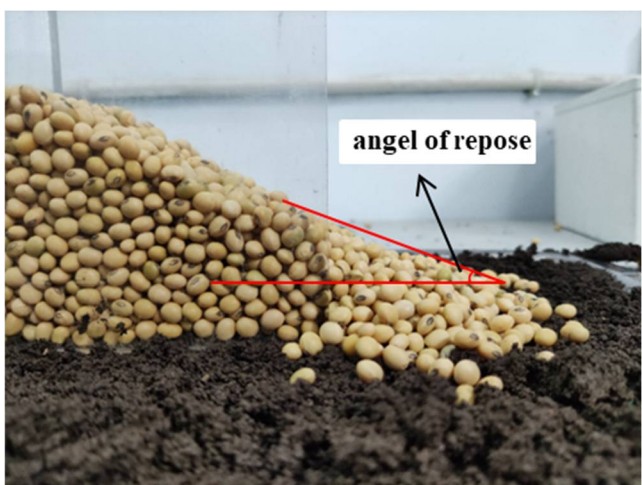

**Figure 3.** Angle of repose of piling test.

### 3.2. Screening of Sensitive Parameters

Taking JD17 as the research subject, the sensitive parameters were screened by means of a PB test. For the simulations, according to the previous study, the soybean seed particle model was 13-sphere model [16], the soil particle model had a particle size of 1 mm and was generated from a uniform distribution of 3-sphere and spherical particle models. The material parameters were determined in previous studies [16–18]. During the simulation, soil particles and soybean seed particles were first generated. After both the soil particles and soybean seed particles had settled, a vertical velocity of 1 m/s was given to the component (wall) corresponding to the insert plate. Then the soybean seed particles flowed out of the loading box.

The remaining parameters were the interaction parameters between the soil and soybean particles, which were as follows; contact pull-off force ($f$), surface energy ($\gamma$), contact plastic ratio ($\lambda$), slope exp ($n_0$), tensile exp ($n$), tangential stiff multiplier ($k$), static friction coefficient ($\mu_1$), rolling friction coefficient ($\mu_2$) and restitution coefficient ($e$).

The slope exp was taken as 1.5 because of the adhesion between the soybean seed particles and the soil particles. The contact pull-off force was set to 0. The rest of the parameter settings are shown in Table 2. Simulations were performed according to the parameters in Table 1. The response index was the angle of repose.

**Table 2.** PB design.

| Standard Order | Run Order | $\gamma$ (J/m²) | $\lambda$ | $n$ | $k$ | $e$ | $\mu_1$ | $\mu_2$ |
|---|---|---|---|---|---|---|---|---|
| 12 | 1 | 0.01 | 0.3 | 1 | 0.5 | 0.1 | 0.1 | 0.01 |
| 5 | 2 | 1 | 0.8 | 1 | 1 | 0.8 | 0.1 | 0.5 |
| 9 | 3 | 0.01 | 0.3 | 1 | 1 | 0.8 | 0.7 | 0.01 |
| 11 | 4 | 0.01 | 0.8 | 1 | 0.5 | 0.1 | 0.7 | 0.5 |
| 7 | 5 | 0.01 | 0.8 | 5 | 1 | 0.1 | 0.7 | 0.5 |
| 2 | 6 | 1 | 0.8 | 1 | 1 | 0.1 | 0.1 | 0.01 |
| 8 | 7 | 0.01 | 0.3 | 5 | 1 | 0.8 | 0.1 | 0.5 |
| 6 | 8 | 1 | 0.8 | 5 | 0.5 | 0.8 | 0.7 | 0.01 |
| 10 | 9 | 1 | 0.3 | 1 | 0.5 | 0.8 | 0.7 | 0.5 |
| 1 | 10 | 1 | 0.3 | 5 | 0.5 | 0.1 | 0.1 | 0.5 |
| 3 | 11 | 0.01 | 0.8 | 5 | 0.5 | 0.8 | 0.1 | 0.01 |
| 4 | 12 | 1 | 0.3 | 5 | 1 | 0.1 | 0.7 | 0.01 |

A screenshot of the simulation results is shown in Figure 4a. The part of the soy bean that flowed out of the loading box in contact with the soil was used as the study object to analyze the angle of repose results, as shown in Figure 4b.

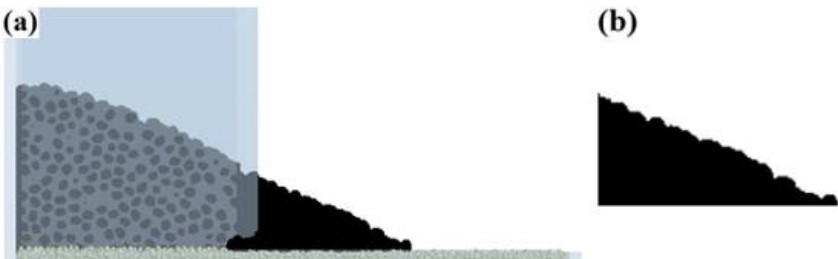

**Figure 4.** (**a**) Graph of simulation results of piling test, and (**b**) angle of repose of soybean seed particles in contact with soil particle.

The results of the PB test were analyzed by Analysis of variance (ANOVA), as shown in Table 3. The linear model had a *p*-value of 0.01, which showed a significant performance. For the different factors, the *p*-value of the coefficient of static friction was 0.001, indicating that it was extremely significant for the response index. The *p*-value of the coefficient of rolling friction was 0.046, indicating that it was significant for the response index. The *p*-values for the other parameters were much greater than 0.05 and were insignificant for the response index. The same conclusion could be drawn from the Pareto diagram of the standardization effect, as shown in Figure 5. Accordingly, the sensitive parameters were determined as the coefficient of static friction and the coefficient of rolling friction.

**Table 3.** Analysis of variance.

| Source | Degree of Freedom | Adj SS | Adj MS | F Value | *p* Value |
|---|---|---|---|---|---|
| Model | 7 | 85.8786 | 12.2684 | 15.37 | 0.010 |
| Linear | 7 | 85.8786 | 12.2684 | 15.37 | 0.010 |
| $\gamma$ (J/m$^2$) | 1 | 0.7197 | 0.7197 | 0.90 | 0.396 |
| $\lambda$ | 1 | 0.0587 | 0.0587 | 0.07 | 0.800 |
| $n$ | 1 | 1.2665 | 1.2665 | 1.59 | 0.276 |
| $k$ | 1 | 0.1518 | 0.1518 | 0.19 | 0.685 |
| $e$ | 1 | 1.7079 | 1.7079 | 2.14 | 0.217 |
| $\mu_1$ | 1 | 75.4841 | 75.4841 | 94.56 | 0.001 |
| $\mu_2$ | 1 | 6.4898 | 6.4898 | 8.13 | 0.046 |
| Error | 4 | 3.1932 | 0.7983 | | |
| Total | 11 | 89.0718 | | | |

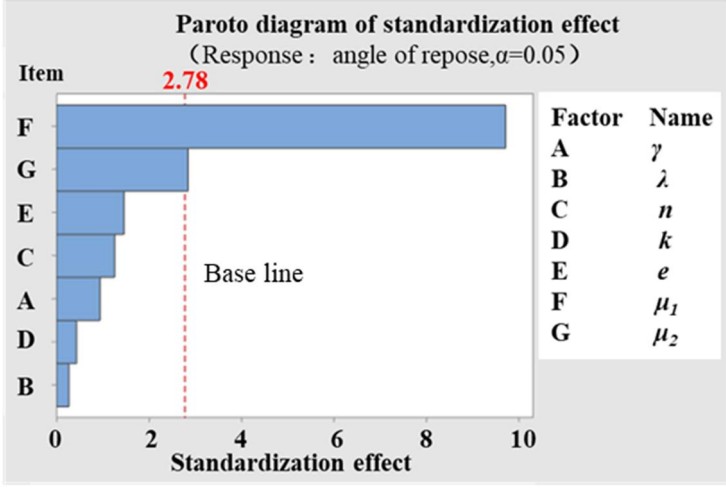

**Figure 5.** Pareto diagram of the standardization effect of PB test.

### 3.3. Calibration of Parameters

A CCD test was performed to calibrate the static and rolling friction coefficients for each variety of soybean seed particles, as shown in Table 4. The other parameters took the system default values.

**Table 4.** CCD test and simulation results of three varieties.

| Standard Order | Run Order | $\mu_1$ | $\mu_2$ | Angle of Repose, Deg | | |
|---|---|---|---|---|---|---|
| | | | | SN42 | JD17 | ZD39 |
| 11 | 1 | 0.355 | 0.255 | 21.20 | 19.70 | 22.18 |
| 10 | 2 | 0.355 | 0.255 | 20.36 | 21.45 | 21.54 |
| 3 | 3 | 0.111 | 0.428 | 19.29 | 20.72 | 21.01 |
| 6 | 4 | 0.7 | 0.255 | 21.99 | 21.57 | 21.78 |
| 2 | 5 | 0.599 | 0.082 | 20.04 | 20.40 | 21.46 |
| 9 | 6 | 0.355 | 0.255 | 20.60 | 22.04 | 21.60 |
| 4 | 7 | 0.599 | 0.428 | 21.51 | 22.25 | 22.03 |
| 5 | 8 | 0.01 | 0.255 | 19.72 | 19.89 | 19.58 |
| 8 | 9 | 0.355 | 0.5 | 1.70 | 21.70 | 22.03 |
| 12 | 10 | 0.355 | 0.255 | 20.81 | 21.42 | 21.47 |
| 7 | 11 | 0.355 | 0.01 | 20.37 | 18.40 | 22.09 |
| 13 | 12 | 0.355 | 0.255 | 20.49 | 21.82 | 21.60 |
| 1 | 13 | 0.111 | 0.082 | 19.16 | 20.53 | 20.49 |

Simulation tests were performed according to the parameters of Table 4. The tests were repeated three times for each variety and the angle of repose data was entered into Table 4. Subsequently, response surface regression analysis was carried out for the three varieties.

For SN42, the ANOVA results are shown in Table 5. For the overall model, the *p*-value was 0.038 and it could be said that the predictive model for the test was significant. The results of the linear analysis showed that the *p*-value for linearity was 0.009, which showed a very significant effect on the test response index. The *p*-value for the static friction coefficient was 0.006, which had a significant effect on the response index. The *p*-values corresponding to the Square and Two-factor interaction were both greater than 0.05, and were not significant for the response index.

**Table 5.** Analysis of variance for SN42.

| Source | Degree of Freedom | Adj SS | Adj MS | F Value | *p* Value |
|---|---|---|---|---|---|
| Model | 5 | 7.11132 | 1.42226 | 4.46 | 0.038 |
| Linear | 2 | 6.47862 | 3.23931 | 10.16 | 0.009 |
| $\mu_1$ | 1 | 4.96897 | 4.96897 | 15.59 | 0.006 |
| $\mu_2$ | 1 | 1.50965 | 1.50965 | 4.74 | 0.066 |
| Square | 2 | 0.18080 | 0.09040 | 0.28 | 0.761 |
| $\mu_1 \times \mu_1$ | 1 | 0.16474 | 0.16474 | 0.52 | 0.495 |
| $\mu_2 \times \mu_2$ | 1 | 0.03189 | 0.03189 | 0.10 | 0.761 |
| Two-factor interaction | 1 | 0.45190 | 0.45190 | 1.42 | 0.273 |
| $\mu_1 \times \mu_2$ | 1 | 0.45190 | 0.45190 | 1.42 | 0.273 |
| Error | 7 | 2.23074 | 0.31868 | | |
| Lack of fit | 3 | 1.79733 | 0.59911 | 5.53 | 0.066 |
| Pure error | 4 | 0.43341 | 0.10835 | | |
| Total | 12 | 9.34206 | | | |

The regression equation was obtained as shown below:

$$\theta = 19.15 + 3.04\mu_1 + 0.83\mu_2 - 2.59\mu_1\mu_1 - 2.26\mu_2\mu_2 + 7.95\mu_1\mu_2 \tag{13}$$

where $\theta$ is the angle of repose; $\mu_1$ and $\mu_2$ are the coefficient of static friction and rolling friction, respectively.

The angle of repose (19.79°) between SN42 and soil particles was used as the optimization target value. Response optimization was performed and the results are shown in Table 6.

**Table 6.** Parameter optimization results of SN42.

| Solve | A | B | Fitting Value of Repose Angle, Deg | Compound Arbitrariness |
|-------|-----|-----|------------------------------------|------------------------|
| 1 | 0.25366 | 0.011 | 19.79 | 1 |
| 2 | 0.116491 | 0.5 | 19.79 | 1 |
| 3 | 0.196487 | 0.037595 | 19.7387 | 0.918 |
| 4 | 0.7 | 0.011 | 20.0846 | 0.86635 |

Using the same simulation and analysis method, the optimization results for JD17 and ZD39 were obtained and are shown in Tables 7 and 8, respectively.

**Table 7.** Parameter optimization results of JD17.

| Solve | A | B | Fitting Value of Repose Angle, Deg | Compound Arbitrariness |
|-------|-----|-----|------------------------------------|------------------------|
| 1 | 0.3555 | 0.124861 | 20.33 | 1 |
| 2 | 0.682333 | 0.049765 | 20.3299 | 0.99993 |
| 3 | 0.682333 | 0.049765 | 20.3299 | 0.99993 |
| 4 | 0.365543 | 0.061835 | 20.1425 | 0.83916 |

**Table 8.** Parameter optimization results of ZD39.

| Solve | A | B | Fitting Value of Repose Angle, Deg | Compound Arbitrariness |
|-------|-----|-----|------------------------------------|------------------------|
| 1 | 0.572115 | 0.183791 | 20.96 | 1 |
| 2 | 0.3555 | 0.373003 | 20.96 | 1 |
| 3 | 0.636982 | 0.165984 | 20.9581 | 0.99893 |
| 4 | 0.562775 | 0.182738 | 20.9407 | 0.98923 |
| 5 | 0.562775 | 0.182738 | 20.9407 | 0.98923 |

For each variety the first set of solutions of the optimization parameters were taken as 0.254 and 0.011 for SN42, 0.355 and 0.125 for JD17 and 0.572 and 0.184 for ZD39.

*3.4. Validation of Calibration Parameters*

Simulations of the piling test were performed to verify the accuracy of the calibrated parameters. The simulation was compared with the test results, as shown in Figure 6. With the previously calibrated parameters, the piling test simulations for SN42, JD17 and ZD39 were all within the error band of the test results, with a difference of 0.68°, 0.9° and 0.05°, respectively, so that the calibrated parameters were accurate.

The simulation parameters between the particles in this paper are summarized in Table 9.

**Table 9.** Summary of simulation parameters between soybean and soil particles.

| | SN42-Soil | JD17-Soil | ZD39-Soil |
|---|-----------|-----------|-----------|
| $e$ | 0.75 | 0.75 | 0.75 |
| $\mu_1$ | 0.254 | 0.355 | 0.572 |
| $\mu_2$ | 0.011 | 0.125 | 0.184 |
| F, N | 0 | 0 | 0 |
| $\gamma$, J/m$^2$ | 0.5 | 0.5 | 0.5 |
| $\lambda$ | 0.35 | 0.35 | 0.35 |
| $n_0$ | 1.5 | 1.5 | 1.5 |
| $n$ | 1 | 1 | 1 |
| $k$ | 0.67 | 0.67 | 0.67 |

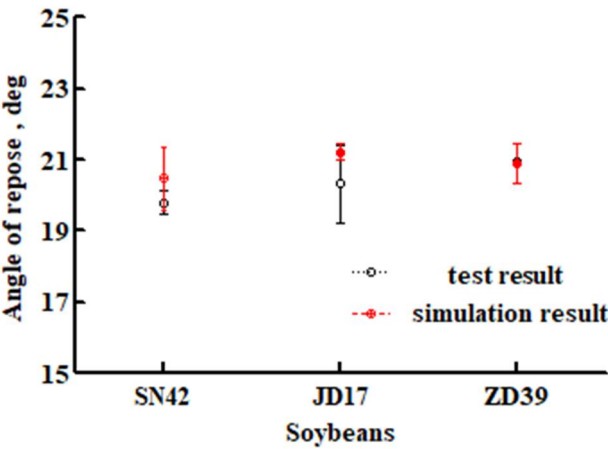

**Figure 6.** Comparison of simulation and test results of angle of repose.

## 4. Seed Throwing Test and Simulation

### 4.1. Seed Throwing Test

The process of collision between soybean seed particles and soil particles during the seed throwing test was very transitory. The bouncing and rolling of the soybean could not be observed by the eye. In this paper, Xu's test setup [10] was improved. Two high-speed cameras were used, which were placed vertically, as shown in Figure 7a. The motion of the soybean seed particles could be captured in both the normal and tangential directions at the same time, as shown in Figure 7b,c.

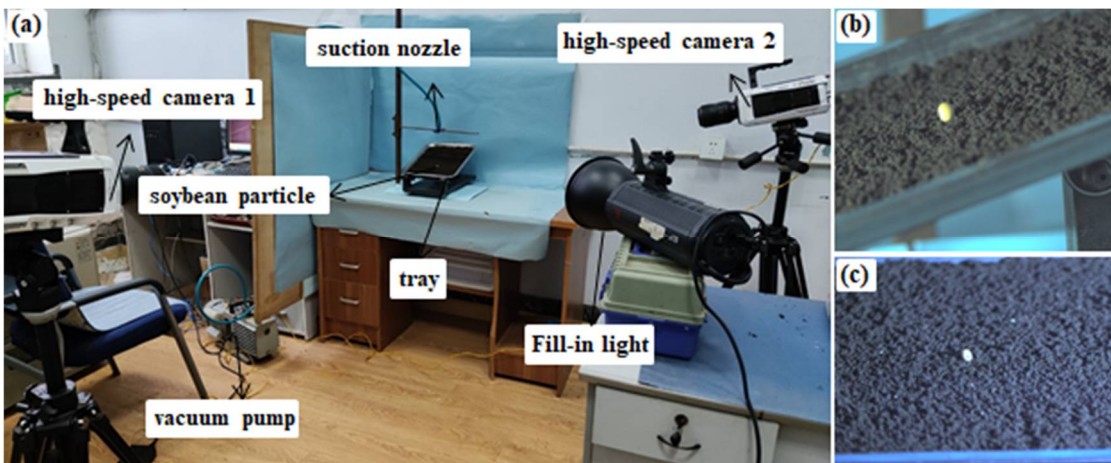

**Figure 7.** (**a**) Seed throwing test apparatus, the movement of soybean seed particles taken by (**b**) high-speed camera 1 and (**c**) high-speed camera 2.

The test procedure was as follows: The vacuum pump and high-speed camera were turned on. The soybean seed particle was attached to the vacuum nozzle. The vacuum pump was disconnected and the soybean seed particle dropped onto the soil surface by gravity, bouncing and rolling. The collision was recorded using the high-speed camera. The throwing height, soil plane inclination angle and collision orientation of the throwing test varied and the test was replicated five times for the three varieties.

### 4.2. Seed Throwing Simulation

The simulation process for the seed throwing test was as follows. Soil particles were generated in the soil tray through the particle factory. After the soil particles settled, a soybean seed particle was generated on top of the soil tray. The screenshot of the simulation of the seed throwing test is shown in Figure 8. The throwing height, soil plane inclination

angle and collision orientation of the throwing simulation varied and the simulation was replicated five times for the three varieties.

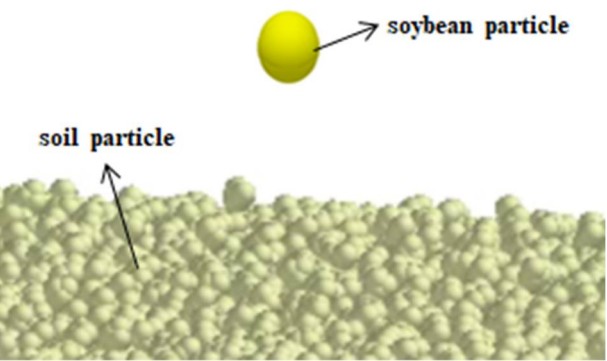

**Figure 8.** Screenshot of seed throwing test simulation.

## 5. Analysis of Results

It was found that the bouncing distances of the soybean seed particles were very small under different circumstances within the scope of this paper, which are analyzed in detail below. As a consequence, the comparison between simulation and test was not carried out for the bounce distance.

### 5.1. The Effect of Seed Throwing Height on Bouncing and Rolling Distance

Figure 9 is the relationship between bounce distance and throwing height for the three varieties. For each variety, the bounce distance was not significant. ZD39 had a maximum bounce distance of 1.85 mm at a throwing height of 150 mm.

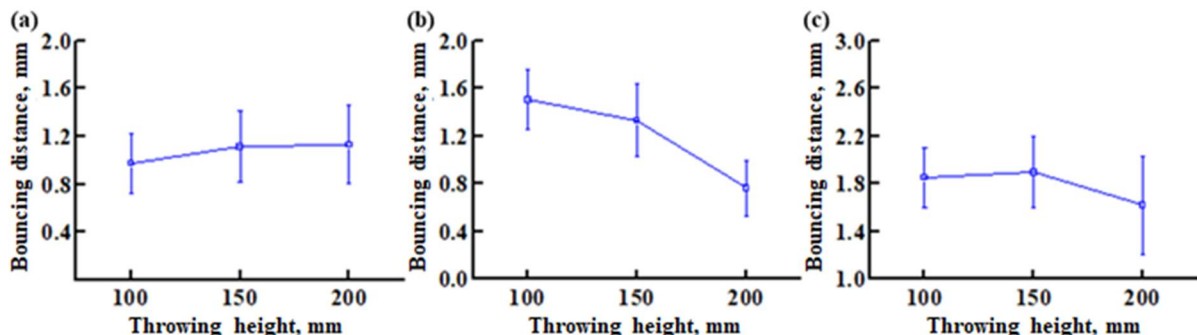

**Figure 9.** The relationship between bounce distance and throwing height for (**a**) SN42 (**b**) JD17 and (**c**) ZD39.

Within the scope of this paper, the variation in bouncing distances of different soybean seed particles varied with increasing seed throwing height. However, the differences between bouncing distances were not significant.

Figure 10 shows the relationship between rolling distance and throwing height for the three varieties. For SN42, the rolling distance gradually increased when the throwing height varied from 100–200 mm, as shown in Figure 10a. For JD17 and ZD39, the test results showed that the rolling distance of soybean did not vary much when the throwing height varied, and the largest difference between the simulation and test results for JD17 was 4.23 mm at a throwing height of 100 mm. The largest difference between the simulation and test results for ZD39 was 4.23 mm at a throwing height of 150 mm, as shown in Figure 10a,b.

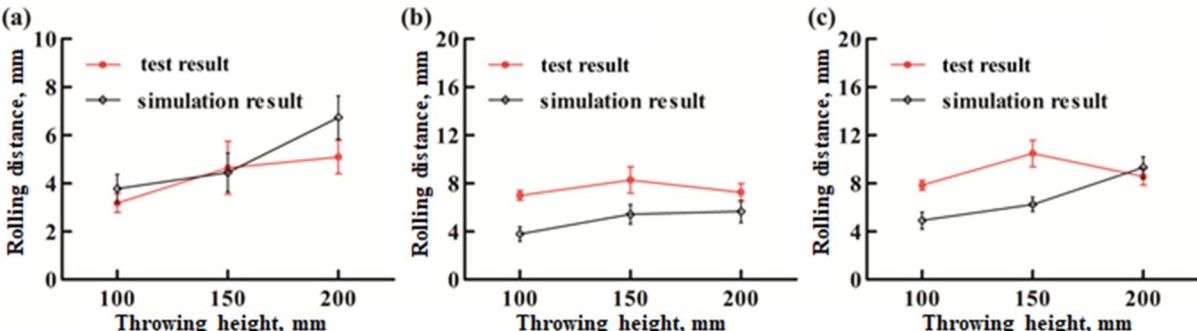

**Figure 10.** The relationship between rolling distance and throwing height for (**a**) SN42, (**b**) JD17 and (**c**) ZD39.

The analysis showed that the bouncing distance of the soybean seed particles was very small when the seed throwing height was varied. However, a certain rolling distance was generated. When the sphericity of the soybean was high, the rolling distance increased as the seed throwing height increased. When the sphericity was low, there was less variation in the rolling distance of the soybean. The trend between the simulation and the test results was essentially the same. Within the scope of this paper, the effect of different seed throwing heights on the bouncing and rolling distance of seeds was not significant, so that any throwing height of 100–200 mm was desirable in practical working conditions.

*5.2. The Effect of Soil Plane Inclination Angle on Bouncing and Rolling Distance*

Figure 11 is the relationship between bounce distance and soil plane inclination angle for the three varieties. For SN42, the bounce distance of soybean seed particles tended to decrease and then increase as the inclination of the soil plane increased, but the overall bounce distance was not significant. For JD17 and ZD39, the bounce distance gradually decreased as the inclination angle of the soil plane increased. At an inclination angle of 0°, the three soybean seed particles had the largest bounce distances, with values of 1.13 mm, 0.76 mm and 1.62 mm, respectively, as shown in Figure 11.

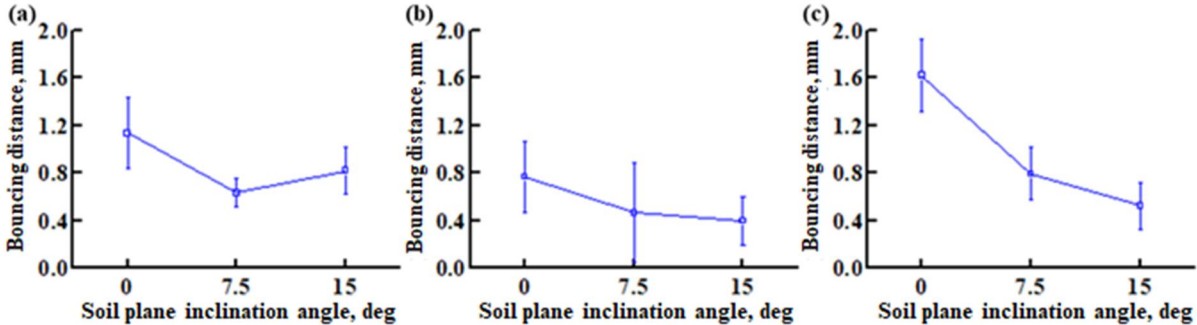

**Figure 11.** Test results of the relationship between bounce distance and soil plane inclination angle for (**a**) SN42, (**b**) JD17 and (**c**) ZD39.

Figure 12 shows the relationship between rolling distance and soil plane inclination angle for the three varieties. From the test results, it can be seen that as the soil plane inclination angle increased, the rolling distance of soybean seed particles became larger. The rolling distances were smallest at 0°, with values of 5.09 mm, 6.25 mm and 8.55 mm, and largest at 15°, with values of 43.16 mm, 25.41 mm and 36.2 mm, respectively. At the same time, the simulation and test results were not very different and had the same trends.

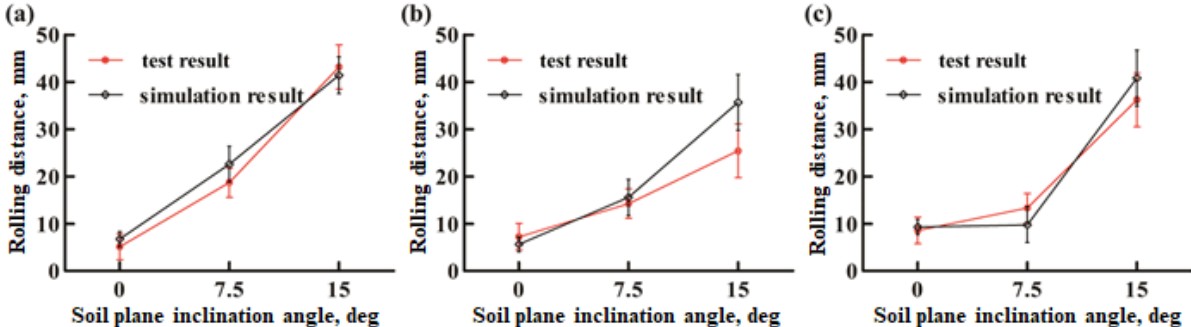

**Figure 12.** The relationship between rolling distance and soil plane inclination angle for (**a**) SN42, (**b**) JD17 and (**c**) ZD39.

The analysis shows that the bouncing distance of soybean seed particles tended to decrease with increasing soil plane inclination angle in the study scope of the test, but the bouncing distance was not large. The rolling distance increased with increasing soil plane inclination angle. In actual sowing, seeds should be placed directly into the furrow instead of the side wall of the seed trench as far as possible, to reduce the bouncing and rolling distance of the seeds.

### 5.3. The Effect of Collision Orientation on Bouncing and Rolling Distance

The soybean seed particles were made to collide with the soil horizontally, laterally and vertically, respectively. When soybean seed particles collided with soil along the direction of T it was a horizontal collision, as in Figure 13a,b. When soybean seed particles collided with the soil along the direction of W it was a lateral collision, as in Figure 13a,c. When soybean seed particles collided with the soil along the direction of L it was a vertical collision, as in Figure 13a,d. The collision process was recorded using the high-speed camera. The simulation could be realized by setting the initial angle of the soybean particle model to collide with the soil particles at different parts.

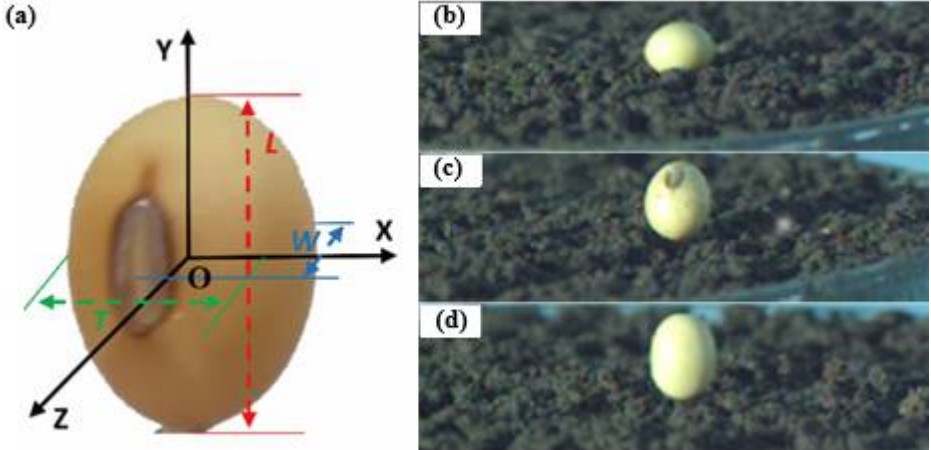

**Figure 13.** (**a**) Tri-axial dimensions of an actual soybean seed particle [16], soybean seed particles collide with soil in different orientations of (**b**) horizontal, (**c**) lateral and (**d**) vertical.

Figure 14 is the relationship between bounce distance and impact orientation for the three varieties. The analysis showed that the bouncing distance of the soybean seed particles varied randomly when soybean seed particles collided with the soil particles in different orientations, but the bouncing distance as a whole was not very high. ZD39 had the highest bounce distance when colliding with soil in the vertical orientation, with a value of 1.62 mm. JD17 had the lowest bounce distance when colliding in the vertical orientation, with a value of 0.76 mm.

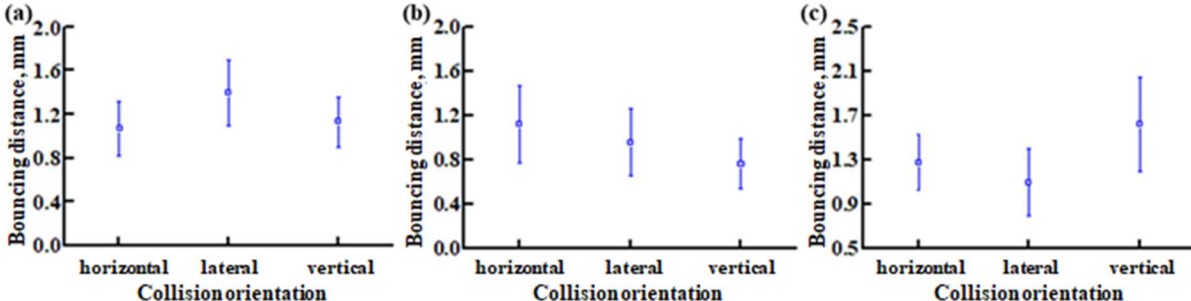

**Figure 14.** The relationship between bounce distance and impact orientation for (**a**) SN42, (**b**) JD17 and (**c**) ZD39.

Figure 15 shows the relationship between rolling distance and collision orientation for the three varieties. For SN42, as shown in Figure 15a, the difference in rolling distance after collision with soil in different orientations was not significant, due to its high sphericity. The rolling distance was the largest for vertical collision and the smallest for lateral collision, with a difference of 1.94 mm. The simulation and test results followed the same trends, but were slightly larger overall than the test results. The largest difference between the test and simulation results was 1.64 mm for the vertical collision.

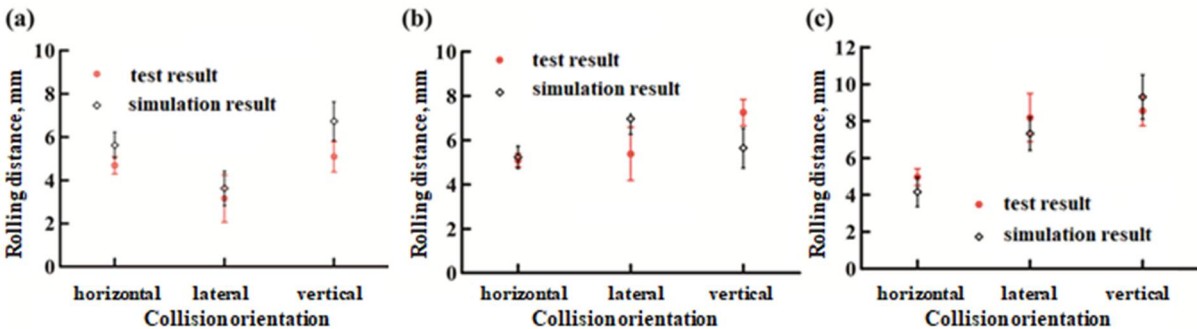

**Figure 15.** The relationship between rolling distance and collision orientation for (**a**) SN42, (**b**) JD17 and (**c**) ZD39.

For JD17, as shown in Figure 15b, the test results showed that the rolling distance was the smallest in the horizontal collision and the largest in the vertical collision, with a difference of 2.16 mm. The difference between the simulation and test results was not significant, with the largest difference being 1.61 mm for the vertical collision.

For ZD39, as shown in Figure 15c, the rolling distance was the smallest in horizontal collisions and the largest in vertical collisions. The simulation and test results were close to each other and all were within the test error.

The analysis showed that different collision orientations had little effect on the bounce distance of the soybean seed particles. However, as regards the rolling distance, if the sphericity of the soybean seed particles was high, the effect of the different collision orientations was not significant. If the sphericity was low, the rolling distance was minimal for horizontal collisions and maximal for vertical collisions. The simulation results were closer to the test results. The greater the sphericity of the seed, the smaller the variation in the rolling distance of the seed particles when they collided in different directions. Therefore, in order to improve sowing accuracy, soybean varieties with greater sphericity should be chosen for the throwing.

*5.4. The Effect of Relative Seed Throwing Speed on Rolling Distance*

A computer vision seeding test bench was used to investigate the effect of relative seed throwing speed on the rolling distance of soybean seed particles. A square soil tray with dimensions of 500 × 500 × 8 mm was processed, as shown in Figure 16. The soil

tray was placed on the seedbed belt and the relative seed throwing speed was varied by changing the speed of the seedbed belt.

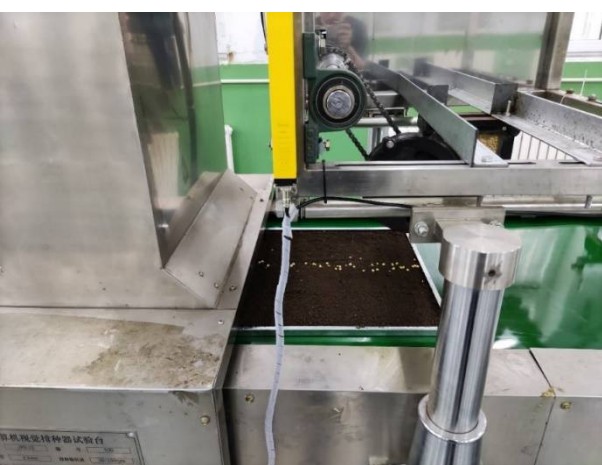

**Figure 16.** Computer vision seeding test bench.

Using the SN42 as an example, the test procedure was as follows: The seeding test bench was turned on, The seeder speed was adjusted to 20 r/min and the seedbed belt speed was adjusted to 0.5 m/s. The soil tray was released from the start of the seed bed belt and the power turned off after completing seed throwing. A camera was used to photograph the distribution of soybean seed particles on the soil tray, as shown in Figure 17.

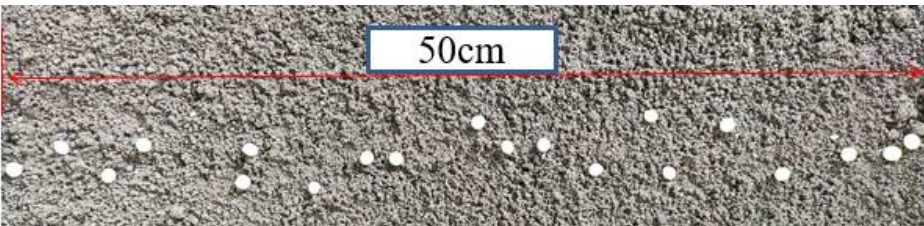

**Figure 17.** Distribution of soybean seed particles on the soil tray.

As a comparison, soybean seeds were thrown directly onto the seedbed belt. Figure 18 shows the distribution of soybean seed particles when seeds were thrown on the seedbed belt and soil tray. In Figure 18a,b there are two red lines, which indicate the line where the soybean seed particles would be if they were not bouncing and rolling.

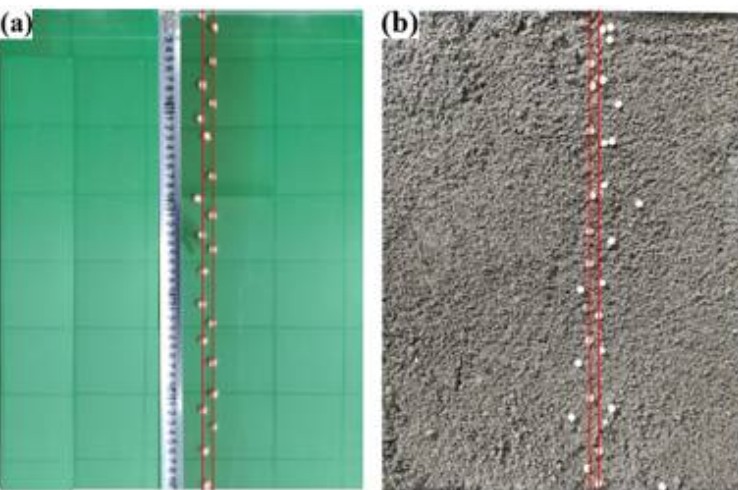

**Figure 18.** Distribution of soybean seed particles on the (**a**) seedbed belt and (**b**) soil tray.

The analysis showed that the soybean seed particles were uniformly distributed on the surface of the seedbed belt, with each row of soybeans in almost the same straight line, as shown in Figure 18a. The distribution of soybean seed particles on the soil tray was much more dispersed, with the number of soybean seed particles deviating from a straight line reaching 36.67%, as shown in Figure 18b. This was also the case for JD17 and ZD39. This phenomenon illustrated the degree of bouncing and rolling of soybean seed particles that occurs when seeds are thrown on the soil surface.

The speed of the seeder wheel during the test was 20 r/min, corresponding to a linear speed of 0.07 m/s. The speeds of the seedbed belt were taken to be 0.5 m/s, 0.75 m/s, 1 m/s, 1.25 m/s and 1.5 m/s. Ignoring air resistance, the relative seed throwing speeds at collision were 0.57 m/s, 0.82 m/s, 1.07 m/s, 1.32 m/s and 1.57 m/s. The test was repeated three times for each variety at different relative seed throwing speeds.

The effect of relative seed throwing speed on seed bouncing and rolling was studied by analyzing the relative width of the distribution of soybean seed particles on the soil tray. To ascertain the distribution of soybean seeds on the soil surface, the two outermost soybean seed particles were found and taken as the reference point. Two straight lines were made along the direction of movement of the soil tray, and the horizontal distance between the two lines was the relative width. The relative widths of the distributions of soybean seed particles at different relative seed throwing speeds are shown in Figure 19a. In order to accurately analyze the relative width between soybean seed particles, the picture was first binarized using image processing software to obtain a binarized map of the seed distribution, as shown in Figure 19b. The relative width of the distribution of soybean seed particles on the seeded belt in the picture was calculated.

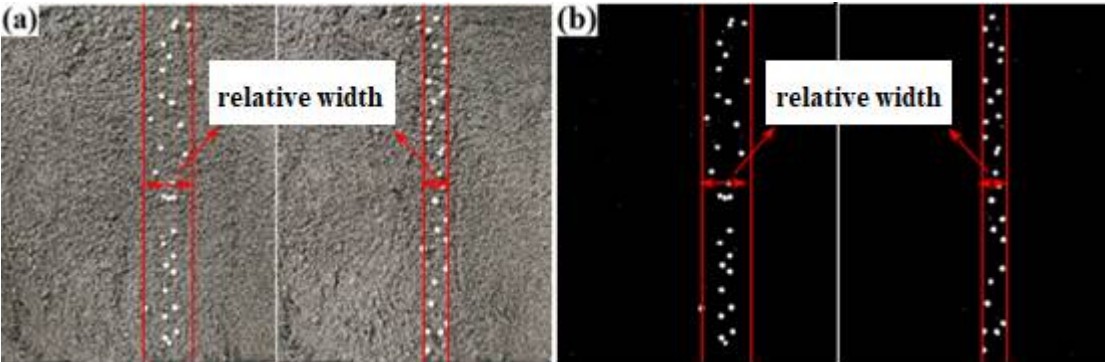

**Figure 19.** Relative width of soybean seed particles distributed on the soil tray at different relative seed throwing speeds (**a**) graph of test results (**b**) picture of binarization of test results.

Simulations were performed to analyze the relative width of soybean seed particles distributed on the soil tray at different relative seed throwing speeds. Figure 20 shows a screenshot of the simulation results for SN42 soybean seed particles at relative seed throwing speeds of 0.57 m/s and 1.57 m/s. A notable difference in the distribution of soybean seed particles could be found with different relative seed throwing speeds.

Figure 21 shows the relationship between the relative width and the relative seed throwing speed. The test results in Figure 21 show that, for SN42 and ZD39, the relative width of the soybean seed sowing strip gradually increased as the relative seed throwing speed increased, meaning that the rolling distance of the soybean seed particles increased as the relative seed throwing speed increased. For JD17, at a relative seed throwing speed of 1.32 m/s, there was a fluctuation and the relative width decreased slightly, but, overall, it was still consistent with the trend that the relative width of the sowing strip of soybean seeds gradually became larger as the relative seed throwing speed increased. For the simulation results, the trend was the same, on the whole, for the three varieties as for the test results. The relative width of the sowing strip increased as the relative seed throwing speed increased. The analysis showed that within the scope of this paper, there was a

general rule that the greater the relative seed throwing speed, the greater the rolling distance of the soybean seed particles. Therefore, to reduce the bouncing and rolling distance of the seeds, it is necessary to reduce the relative seed throwing speed appropriately.

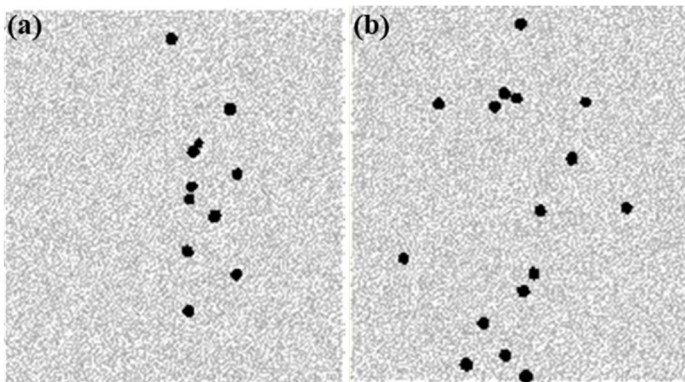

**Figure 20.** Simulation results of the distribution of soybean seed particles on the soil tray at relative seed throwing speeds of (**a**) 0.57 m/s and (**b**) 1.57 m/s.

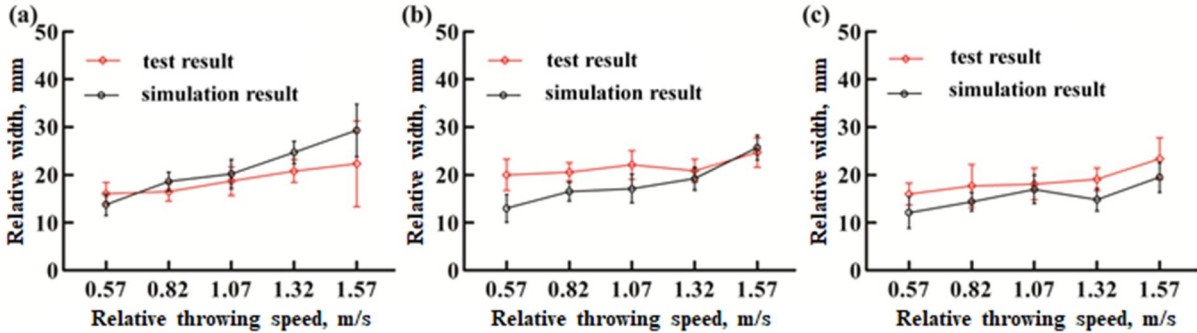

**Figure 21.** The relationship between relative width and relative seed throwing speed for (**a**) SN42, (**b**) JD17 and (**c**) ZD39.

## 6. Conclusions

This paper is the first study of the seed throwing process of soybean seeds. The effect of different factors on the bouncing and rolling distance of soybean seed particles during seed throwing was analyzed in detail using a high-speed camera test setup. Theoretical support was provided for the realization of precision seeding. The simulation analysis of the seed throwing test was carried out. The accuracy of the contact model selection was verified. A parameter calibration method between soybean seed particles and soil particles is proposed. It provides a reference for scholars to study the simulation analysis of seed throwing for different seeds. The main conclusions are as follows:

(1) The presence of cohesion between soybean seed particles and soil particles was demonstrated by a texture test. It proved that the choice of EEPA model is accurate.

(2) The parameters between soybean seed particles and soil particles were calibrated by a piling test. The static and rolling friction coefficients were identified as sensitive factors through PB test. The optimized parameter values were determined by CCD test. The accuracy of the calibrated parameters was proven by comparing the simulation and test results of the piling angle.

(3) Within the scope of this paper, the bouncing distances of soybean seed particles were all small. For the rolling distance, the relationship between seed throwing height and rolling distance had a certain randomness. The greater the soil plane inclination angle the greater the rolling distance. When the sphericity of the soybean seed particles was high, the effect of different collision orientations was not obvious. If their sphericity was low, the rolling distance was shortest when colliding in the horizontal orientation

and longest when colliding in the vertical orientation. The greater the relative seed throwing speed, the greater the rolling distance of the soybean seed particles.

(4) The simulation of the seed throwing process was performed. Comparing the simulation and test results showed that the trends between the simulation and test results were generally consistent. It was demonstrated that the analysis of the seed throwing process using DEM simulation was accurate and feasible, and proved once again that the calibrated parameters were accurate. It provides a reference for researchers to simulate different seed throwing processes.

**Author Contributions:** Conceptualization, D.Y.; methodology, D.Y.; validation, D.Y. and L.W.; investigation, D.Y. and; resources, J.Y.; writing—original draft preparation D.Y.; writing—review and editing, N.Z.; supervision, Y.T.; project administration, L.W.; funding acquisition, J.Y. All authors have read and agreed to the published version of the manuscript.

**Funding:** The authors are grateful to the National Natural Science Foundation of China (No. 52130001) for the financial support of this work.

**Institutional Review Board Statement:** Not applicable.

**Informed Consent Statement:** Not applicable.

**Data Availability Statement:** Not applicable.

**Conflicts of Interest:** The authors declare no conflict of interest.

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
