# Peer review of "Test and Simulation Analysis of Soybean Seed Throwing Process"

_processes, doi:10.3390/pr10091731_

Round 1

Reviewer 1 Report

33 Through experimental tests

34 “where made”. I don’t understand the sense of the period.

37 if you cite the authors like in line 32, is better place the reference near the name of the authors.

41 what do you mean by computer simulation? the application of the model? explain better

47 what are the two different particles you mention? explain better.

77 report the names and characteristics of the three types of seeds. The reader can better follow the following discussion

90 same problem of 47.

91 “from previous research”. Please use the citations to indicate the studies referred to. For example, “as seen in [citation] the model used in the present work is Edinburgh Elasto-Plastic Adhesion (EEPA), capable to the describe the soil particles interactions”

95 a repetition for “very sticky soil”. Check the phrase

98 is the first time that use JD17, is better if you introduce it before

113 113 is okay to retrieve the study but is helpful for the reader at least to have a summary table with the parameters used. Maybe is better place here the part of the table 8 that are already know before the calibration.

118 the level of soil moisture is an important parameter that gives different mechanical characteristics to the soil. How was this value taken? Is the same for all the tests? If it is the typical or optimal value of the soil for sowing it is better to specify it

127 you mentioned the 13-sfere model. Are there any tests to understand the goodness of using this model? instead, if the choice comes from literature studies it is right to mention them. It helps the reader to understand the reason of the choice. Would be helpful to insert an image with the seed image and its DEM representation to face a comparison.

The particle soil is of 1 mm, is bigger than the real one? And if is so, motivate your choice.

The definition of the domain is missing: is it equivalent to the experimental apparatus? how large is it? what are the boundary conditions?

Furthermore, since the simulation is time-dependent, the initial configuration must be declared:

Are soil particles allowed to settle first?

What is the initial configuration of the particles’ block representative of the seeds?

In the simulation how is the wall removed to let the seeds fall?

Please integrate the missing data. The simulations must be clear and replicable.

130 it is okay to retrieve the study, but it helps the reader at least to have a summary table with the parameters used

221 is important to define the measurement of the results: the bounce distance is the maximum height reached by the seed after the first bounce or the maximum reached before it finally stops? Is the rolling distance the maximum distance measured from the point of impact or just when does it start rolling?

Figure 9 - Decrease the vertical axis to better show the trend of the graph points

245 you mentioned the sphericity, but how much is its value and how much does it change passing from one species to another? Furthermore, rolling friction also changes from one species to another, do you think it has an effect?

figure 11 - Decrease the vertical axis to better show the trend of the graph points

260 formatting error

274 better explain the three types of orientation. The figure helps, but it can be improved

figure 14 - Decrease the vertical axis to better show the trend of the graph points

figure 18 – is possible to have a picture without the photographer?

346 – explains how relative widths are measured: in the maximum horizontal distance?

356 – also here, in which manner do you set your simulation?

412 – I think is a misprint

In a lot of figure, in the axis label there is “name of variable , measure unit”, please correct the space before the comma

Reviewer 2 Report

The manuscript discusses the calibration of DEM model for the interaction between soybean seed particles and soil particles, and then the soybean throwing process. This is an interesting application of particle simulation in agriculture. However, there are the following issues in the current version:

1. The originality is not clearly introduced and summarised. The calibration of the model is like a routine job in simulation, which looks not new but maybe just the particles are changed. In conclusion point (4), it is said parameters are from previous research, and the study "again" proves the parameters are correct. This conclusion point shows limited originality.

2. The connection between the studied process and the practical application should be given in more detail. The introduction should discuss why this process is worth research (for the real applications), and what kind of application problems can be solved in this research. In the analysis, the results should be compared from the application point of view, e.g., which results in Fig. 20 and Fig. 21 are preferred in the application.  In the conclusion, some recommendations should be given. 

3. In the calibration part, the calibrated parameters should be defined. The contact force model should also be given as the parameters are used in the equations. 

4. Moderate editing is needed.

Round 2

Reviewer 1 Report

Good job! 

just two minimum adjustments:

11 round brackets with no content inside

192 "according to the previous study", I presume.

Author Response

Response 1:According to the reviewer's comment,the author deleted the brackets,as detailed in line 11.

Response 1:According to the reviewer's comment,the author revised the word,as detailed in line 192.

Reviewer 2 Report

The revised manuscript can be recommended.

Author Response

According to the comment of reviewer, the author carefully checks the format and grammar in the paper.